# Comparison of Acute Arterial Responses Following a Rescue Simulation and Maximal Exercise in Professional Firefighters

**DOI:** 10.3390/healthcare11071032

**Published:** 2023-04-04

**Authors:** Vanessa Santos, Luís Miguel Massuça, Luís Monteiro, Vítor Angarten, Mark G. Abel, Bo Fernhall, Helena Santa-Clara

**Affiliations:** 1Exercise and Health Laboratory, CIPER, Faculdade de Motricidade Humana, Universidade de Lisboa, Cruz Quebrada, 1649-004 Lisboa, Portugal; 2Kinesio Lab, Research Unit in Human Movement Analysis, Instituto Piaget, 2805-059 Almada, Portugal; 3First Responder Research Laboratory, University of Kentucky, Lexington, KY 40506, USA; 4CIDEFES, Lusófona University, 1749-024 Lisbon, Portugal; 5ICPOL Research Center, Higher Institute of Police Sciences and Internal Security, 1300-352 Lisbon, Portugal; 6College of Nursing and Health Sciences, University of Massachusetts, 100 Morrissey Boulevard, Boston, MA 02125, USA

**Keywords:** arterial stiffness, pulse wave velocity, rescue simulation, firefighter

## Abstract

Cardiovascular events are the leading cause of on-duty deaths among firefighters. Screening firefighters for risk of sudden cardiac event is a critical element of a comprehensive medical program. Although intense physical exertion has been shown to trigger sudden cardiac events in the general population, it is unclear how hemodynamic responses following clinical exercise testing compare to that of performing firefighting tasks in personal protective equipment. Therefore, the purpose of this study was to compare hemodynamic responses following rescue simulation (RS) and maximal exercise in firefighters. This was a cross-over repeated measures study. Thirty-eight professional firefighters (31.8 ± 5.2 yr; *V*O_2peak_: 57.9 mL/kg/min) completed a maximal aerobic exercise test (MAET) and an RS. Pulse wave velocity (PWV), pulse pressure (PP), and brachial and central mean arterial pressure (MAP) were measured before and 5 and 15 min post-exercise. The findings indicated that femoral PWV decreased after MAET and RS at both time points (*p* < 0.005). No significant differences were found in aortic and carotid PWV over time or between conditions (*p* ≥ 0.05). Significant increases in brachial and central PP and MAP were noted 5 min post-MAET and RS (*p* = 0.004). In conclusion, the present study demonstrated that peripheral arterial stiffness (AS) decreased in firefighters following both conditions, with no differences in central AS. Our findings provide valuable information on hemodynamic responses similar between RS and MAET, and are important for controlling CVD risk and the AS response.

## 1. Introduction

Sudden cardiac events are the leading cause of on-duty deaths among firefighters [1]. Firefighters tend to have an increased number of cardiovascular disease (CVD) risk factors compared to the general population [2,3,4]. Cardiovascular risk factors have an unfavorable impact on arterial function, including decreased endothelial function, and increased arterial stiffness (AS) [5] and central blood pressure (BP) [6]. These arterial function factors have been found to be predictive of all-cause cardiovascular mortality [7,8].

The incidence of sudden cardiac events is likely elevated among firefighters due to the intense physical demands, psychological stress, and thermoregulatory challenges imposed by an emergency operation. These factors lead to profound sympathetic activation that manifests, in part, with elevated blood pressure and maximal or near-maximal heart rates [9]. Intense physical exertion has independently been shown to trigger sudden cardiovascular events, particularly in those who are sedentary [10,11]. In firefighters, the risk of cardiac event increases due to performing high intensity occupational tasks in fully encapsulating personal protective equipment which collectively increases core temperature and produces hypohydration [12,13]. This physiological sequela produces cardiovascular and thermal strain, which increases thrombotic potential [9,11]. Firefighting activities also produce cardiac fatigue, coupled with a decrease in systemic arterial compliance [14]. Furthermore, emotional stress and physical work near maximal capacity have been found to produce changes in increased hemodynamics and arterial function [15]. Collectively, these factors place predisposed firefighters at risk for a sudden cardiac event.

Despite existing research describing the hemodynamic responses following exercise in a clinical setting among non-firefighter populations, there is a lack of data describing post-exercise hemodynamic responses [6,7,8,10,15]. This information will provide researchers, clinicians, and practitioners with an understanding of firefighters’ hemodynamic responses after performing intense occupational tasks in personal protective equipment. Furthermore, there is limited research providing a comparison of hemodynamic responses following fire rescue operations and maximal exercise in a single cohort of firefighters. This information will guide the development of appropriate clinical screening assessments and countermeasures to reduce firefighters’ cardiovascular risk. Therefore, the purpose of this study was to compare the arterial stiffness and hemodynamic responses of firefighters following simulated fire rescue operations and maximal aerobic exercise test (MAET). We hypothesized that AS and hemodynamics would be similar following fire rescue operations and maximal aerobic exercise.

## 2. Materials and Methods

### 2.1. Subjects

Six hundred and seven professional firefighters were assessed for cardiovascular risk and physical fitness. From this sample, 44 male firefighters qualified to participate in the study based on meeting the inclusion criteria of achieving a positive classification on the shuttle run test (four positive stages: excellent, very good, good, or reasonable) and being <45 years of age. Of those 44, 11 firefighters were selected from each positive rating (excellent, very good, good, and reasonable). However, six firefighters did not complete both RS and MAET assessment sessions (due to incompatibility of schedules), therefore a total of 38 firefighters had complete data for both sessions and were included in the study. All firefighters had experience in performing search and rescue operations. All firefighters were fully informed of the purpose of the study and provided written informed consent. The Faculty of Human Kinetics Ethics Committee approved this study (N28/2017), and all procedures and treatment of subjects were in accordance with the Declaration of Helsinki.

### 2.2. Study Design

The present study utilized a cross-over design with repeated measures to compare AS outcomes from MAET and RS conditions. The independent variables were time (rest vs. post-activity) and condition, whereas the dependent variables were AS outcomes. Participants attended two separate sessions consisting of one treadmill MAET and an RS similar to typical fireground operations. There was a minimum of one week between sessions and a maximum of three weeks.

All sessions were conducted in the afternoon between 2:00 and 5:00 p.m., with each firefighter performing sessions at the same time of day to minimize any potential diurnal variation. Firefighters were instructed to avoid alcohol, caffeine, and vigorous exercise for at least 24 h preceding each assessment session, and firefighters fasted 2–4 h prior to the testing session. In addition, cardiovascular risk was assessed in this sample with the Framingham score [16].

### 2.3. Test Sessions

Firefighters were assessed at rest after spending 15 min in the supine position in a quiet, dimly lit temperature-controlled laboratory, and at 5 and 15 min following the MAET and RS trials. The MAET and RS sessions were 12 to 15 min in duration. HR was continuously monitored (Polar H7 monitor, Kempele, Finland) and Rating of Perceived Exertion (RPE) was assessed [17].

### 2.4. Maximal Aerobic Exercise Testing

Maximal aerobic exercise testing was performed on a treadmill with an individualized incremental protocol based on the maximal velocity reached on the 20 m shuttle run test. The protocol started with a 5 min warm-up at the maximal velocity reached on the shuttle run test. The shuttle run test has been selected as one of the most commonly used tests to assess cardiorespiratory capacity, as shown by Massuça et al. [18]. The velocity was increased by 26.8 m/min every 2 min for 4 min, after which treadmill grade was increased by 2.5% every minute until volitional exhaustion. The protocol ended with a 1 min active recovery at 66.6 m/min plus 2 min of passive recovery in a seated position. RPE was assessed each minute using a 6–20 category-ratio scale [19]. BP was recorded at baseline, during each exercise stage, at peak exercise and each minute during recovery. BP was assessed by a certified physician via auscultation using an aneroid sphygmomanometer.

The test was completed when firefighters reached volitional exhaustion or at least three of the following criteria were achieved: (i) a peak HR greater than 90% of the maximal HR predicted (220-age); (ii) plateau in oxygen consumption (*V*O_2_; i.e., <150 mL/min or <2.1 mL/kg/min increase in *V*O_2_ with concurrent increase in work rate); (iii) a respiratory exchange ratio of >1.1; (iv) a RPE between 18–20 [20].

Inspired and expired gases were continuously analyzed, breath-by-breath, with a portable gas analyzer (K4b2, Cosmed, Rome, Italy). The portable gas analyzer was calibrated prior to each testing session using ambient air and standard calibration gases of known concentration (16.7% O_2_ and 5.7% CO_2_). The calibration of the turbine flowmeter was performed using a 3.0 l syringe (Quinton Instruments, Seattle, WA, USA). Peak oxygen uptake (*V*O_2peak_) was defined as the highest 20 s value attained during the MAET.

### 2.5. Rescue Simulation

The RS was performed in normal ambient temperatures and consisted of activities similar to those performed during an actual rescue operation. Firefighters wore full firefighting EPI (mass, 25 kg) and self-contained breathing apparatus (SCBA) while carrying a 2.5-inch hose (mass, 20 kg). Specifically, firefighters were asked to perform the following tasks as quickly as possible: Firefighters initiated the RS from the basement of a structure by carrying mannequin 30 kg manikin up three flights of stairs (10 stairs per flight) out of the building. Next, the participant climbed six flights and dragged an 80 kg manikin to the safe zone on the 6th floor landing (±200-m). Each floor had three flights of 10 stairs each (total steps: 180). No rest was provided between the rescues. Firefighters provided their RPE (6–20 Borg scale) [19] before starting and immediately after the RS.

Inspired and expired gases were continuously analyzed in both exercises, breath-by-breath, through a portable gas analyzer (K4b2, Cosmed, Rome, Italy). During the RS, the self-contained breathing apparatus mask was replaced by the gas analyzer mask. Heart rate was continuously assessed using a chest strap (Polar H7 Electro, Kempele, Finland).

### 2.6. Brachial Blood Pressure

The right brachial systolic BP (SBP) and diastolic BP (DBP) were measured, using an automatic cuff (Omron 907, Tokyo, Japan) following at least 15 min of supine rest prior to both drills. If the difference in systolic BP between the two measurements was larger than 10 mmHg, another BP measurement was performed. The average of the two closest values was used for analysis. The mean arterial pressure (MAP) was calculated with the formula [DBP + 1/3 (SBP − DBP)] and pulse pressure (PP = SBP − DBP).

### 2.7. Central and Peripheral Arterial Stiffness Indices

All AS measurements were taken with the participant in the supine position after 15 min of rest before each session and 5 and 15 min following each exercise session. An ultrasound scanner and linear 13 MHz probe (MyLab One, Esaote, Italy) with Quality AS technology was used to assess AS outcomes. The ultrasound scanner was placed on the right common carotid artery segment ~1 cm before the bifurcation. The right carotid pressure waveform was calibrated to right brachial diastolic and MAP. Carotid pulse wave velocity (PWV) was calculated with the following equation:PWV = 1/√(ρ∙DC) = √((D^2^∙ΔPP)/(ρ∙(2∙D∙ΔD + ΔD^2^)))
where D, diastolic diameter; ΔD, change of diameter in systole; DC, distensibility coefficient; ΔPP, local pulse pressure; and ρ, blood density [19]. PWV was also measured using applanation tonometry immediately after the ultrasound scan. Carotid, femoral, and distal posterior tibial arteries on the right side of the body were located by a single operator who marked the point for capturing the corresponding pressure curves with two specific pressure sensitive transducers. The distance between the carotid, femoral, radial, and distal posterior tibial arteries was measured directly and entered into Complior analysis software using a correction factor of 0.8 (ALAM Medical, Paris, France). Right brachial BP was measured and entered into the software, and then signal acquisition was started. Whenever a continuous drop before the sharp systolic upstroke was not clearly seen or the tolerance was above 0.5 m/s, a third second measurement was taken. The coefficients of variation for repeated measurements in our laboratory for PWV were <1%.

### 2.8. Dual-Energy X-ray Absorptiometry

Dual-energy X-ray absorptiometry was used to characterize body composition. An extended analysis program for body composition (Hologic Explorer-W, fan-beam densitometer, software QDR for windows version 12.4, Waltham, MA, USA) was used to determine whole body fat mass and lean soft tissue. The same technician positioned the participants, performed the scans, and completed the scan analysis according to the operator’s manual using the standard analysis protocol.

### 2.9. Statistical Analysis

Basic statistics (mean ± standard deviation) were used to describe measures of central tendency and dispersion. Variable distributions were assessed for normality using a Shapiro–Wilk test. Cardiorespiratory and hemodynamic variables were compared between MAET and RS conditions using paired-sample *t*-tests. The analyses of interventions (MAET and RS) over time (rest vs. post-intervention time points) for AS indices were examined via two-way ANOVA with two within factors. Paired (dependent)-sample *t*-tests were used for post hoc analysis. Since AS is dependent on arterial pressure [21], percent changes in MAP from rest were included as a covariate. The level of significance was set at *p* < 0.05. Data were analyzed using SPSS version 24.0 for Windows (SPSS Inc., Chicago, IL, USA).

## 3. Results

The firefighters’ characteristics are displayed in Table 1. Table 2 presents the acute cardiorespiratory and hemodynamic variables from MAET and RS conditions. Compared to the MAET condition, *V*O_2peak_, and heart rate at 5 and 15 min post-RS were significantly lower, whereas brachial MAP 5 min post-RS was significantly greater.

Table 3 presents the comparison of AS outcomes at rest and post-activity in MAET and RS conditions. Regarding PWV outcomes, there was a significant main effect of time on femoral PWV (MAET: ƞ^2^ = 0.20, RS: ƞ^2^ = 0.30, *p* < 0.005), as PWV decreased 5 and 15 min post MAET and RS. There were no interaction effects between conditions or main effects in other AS variables.

A significant time effect was found for central and brachial BP variables. Specifically, there were significant increases in both conditions in central MAP (MAET: ƞ^2^ = 0.33, RS: ƞ^2^ = 0.55, *p* < 0.001), central pulse pressure (MAET: ƞ^2^ = 0.46, RS: ƞ^2^ = 0.41, *p* < 0.001), brachial MAP (MAET: ƞ^2^ = 0.37, RS: ƞ^2^ = 0.55, *p* < 0.001) and brachial pulse pressure (MAET: ƞ^2^ = 0.55, RS: ƞ^2^ = 0.70, *p* < 0.001) from rest to 5 min post activity. After 15 min the values returned to baseline (Table 3).

## 4. Discussion

The main finding of this study indicated that AS and hemodynamic outcomes were largely similar between RS and MAET conditions. Thus, a MAET may provide a viable tool to screen predisposed firefighters for being at risk of a sudden cardiac event while performing fire suppression and rescue operations in personal protective equipment. Therefore, fire departments may consider including MAET AS assessments as part of fire personnel medical examinations to identify at-risk firefighters. Despite these informative findings, it is important to note that these results were derived from a sample of young, lean, and aerobically fit firefighters, which displayed favorable AS responses post-MAET and RS. It is important that future research evaluates AS responses among firefighters representing diverse age, body composition, and aerobic fitness profiles, as these characteristics have been shown to negatively impact AS outcomes [22].

In addition, the present study demonstrated that peripheral AS decreased following MAET and RS, while there were no significant changes in central AS. Additionally, decreases in brachial mean arterial pressure were observed following 15 min of victim rescue operations. In the relevant literature, Fahs et al. reported an increase in aortic stiffness following 3 h of live fire suppression activities, even in the presence of peripheral vasodilation, in young healthy firefighters [23]. Similarly, Lane Cordova et al. reported increased aortic stiffness following 18 min of high-intensity live fire suppression activities in middle-aged firefighters [24]. These responses differ from the present study, likely due to differences in the longer duration of these protocols, exposure to higher ambient temperatures, the physical demands of the firefighting tasks, and possibly the age, fitness, and body composition of the participants. Similar to our findings, several studies have demonstrated that peripheral distensibility is improved with high intensity exercises. However, this type of work also produces a marked decrease in central distensibility and an increase in central AS [25,26,27]. These are similar to the arterial response following dynamic muscle strengthening exercise [26,27,28,29].

In the study by Rakobowchuk et al. participants performed supramaximal exercise (4 Wingate tests separated by 4.5 min) [25]. The central PWV increased, indicating a decrease in distensibility immediately after the sessions and returned to baseline resting levels at 20 min of recovery. Peripheral PWV decreased immediately following the exercise bouts, indicating an increase in distensibility after the two exercise sessions and returning to rest levels after 44 min of recovery [25]. Additionally, Rossow et al. determined the cumulative effects of repeated cycling sprints (Wingate tests), indicating that a single cycling sprint reduces carotid artery compliance immediately after exercise, but a second identical bout of cycling reduces carotid artery compliance even further [30]. In that study, the change in carotid stiffness following a second supramaximal exercise bout appeared to be mediated by factors in addition to change in local distending pressure, such as change in heart rate. Thus, it is clear that supramaximal exercise increases both aortic PWV and carotid stiffness showing a general increase in central AS. Conversely, metabolite-induced vasodilatation reduces peripheral stiffness in the exercised limbs [25]. These studies corroborate our results where peripheral AS decreased following our two high-intensity exercises.

Fahs et al. reported an increase in central artery stiffness 3 h after firefighting activities [23]. Similarly, Lane-Codova et al. also reported an increase in AS after 3 h of live firefighting [24]. In contrast to the present study, increases in AS have been observed, but these studies suggested that there are numerous potential mechanisms which may explain the rise in AS following firefighting tasks. The changes in blood pressure during firefighting activities may affect AS. This could be important for controlling CVD risk and the AS response. So, we can understand that high intensity should be implemented during their workout routine as accommodation and as a preventive measure for cardiovascular stress for real life drills. In addition, Santos et al., Hasegawa et al., and Ramirez-Velez et al. have shown recently that HIIT appears to be effective at reducing the stiffness of central arteries [26,27,31,32].

There are several limitations to the present study. For instance, the present study may be limited by a potential selection bias such that all firefighters were relatively young, as age has been found to be a predictor of AS. Furthermore, this sample was aerobically fit. Collectively, the younger age and high aerobic fitness of this sample may have influenced the hemodynamic outcomes and may not be generalizable to older and less aerobically fit firefighters. An additional limitation is the lack of female participants. No females were included in the current study as there were no females enrolled in the selected groups. As such, future research should include larger sample sizes to examine for potential changes in AS of both male and female firefighters of varied ages, fitness, and body composition levels.

## 5. Conclusions

In conclusion, the current study found that the AS and hemodynamic responses following a simulated fire rescue operation and maximal aerobic exercise were largely similar among young, lean, and aerobically fit firefighters. Additionally, peripheral AS decreased, while central AS remained unchanged in both conditions. Moreover, evidence suggests that HIIT may be an effective training method for reducing central arterial stiffness.

The study’s results suggest that further research is necessary to evaluate the usefulness of AS assessments as part of a comprehensive occupational medical program for firefighters to identify individuals at risk for sudden cardiac events. The findings also highlight the importance of continued research to explore the potential benefits of HIIT in improving cardiovascular fitness and reducing the risk of arterial stiffness and related cardiovascular events in firefighters.

## Figures and Tables

**Table 1 healthcare-11-01032-t001:** Characteristics of firefighters (*n* = 38).

	Mean ± SD
Age (year)	31.8 ± 5.2
Body mass (kg)	75.0 ± 7.7
Height (cm)	175.1 ± 5.2
Body Mass Index (kg/m^2^)	24.5 ± 2.2
Fat mass (%)	15.9 ± 4.0
Lean mass (kg)	9.4 ± 1.0
Diabetes (%)	0
Hypertension (%)	0
Hyperlipidemia (%)	0
Overweight/Obesity (%)	0
Smokers (%)	18
Framingham score (%)	<2%

**Table 2 healthcare-11-01032-t002:** Comparison of acute cardiorespiratory and hemodynamic variables in maximal aerobic exercise test and rescue simulation conditions in 38 firefighters.

Variable	MAET	RS	*p*-Value
Duration (min:s)	11:32 ± 1:30	10:13 ± 2:10	N/A
Peak *V*O_2_ (mL/kg/min)	57.9 ± 8.6	51.7 ± 5.4	0.001 *
Brachial MAP at rest (mmHg)	86.0 ± 6.1	88.5 ± 5.6	0.079
Heart rate at rest (b/min)	60.7 ± 10.1	64.5 ± 4.7	0.084
Peak heart rate (b/min)	185.7 ± 10.5	178.7 ± 11.7	0.059
Heart rate at 5 min post (b/min)	98.8 ± 13.5	95.2 ± 14.3	0.001 *
Heart rate at 15 min post (b/min)	93.2 ± 12.0	87.3 ± 14.9	<0.001 *

*, Significantly different between conditions; MAET, maximal aerobic exercise test; RS, rescue simulation; Peak *V*O_2_, Peak oxygen update; MAP, mean arterial pressure; N/A, not applicable.

**Table 3 healthcare-11-01032-t003:** Comparison of arterial stiffness and hemodynamic indices at rest and following maximal aerobic exercise test and a rescue simulation in 38 firefighters.

Variables	Sessions	Rest	5 min Post	15 min Post	*p*-Value ^§^
Central MAP (mmHg)	MAET	86.1 ± 6.0	92.8 ± 11.0 *	87.0 ± 6.3	0.001
	RS	88.4 ± 5.8	98.5 ± 12.1 *	87.0 ± 8.1	<0.001
Central PP (mmHg)	MAET	57.7 ± 18.0	65.3 ± 19.3 *	52.6 ± 13.8	<0.001
	RS	59.4 ± 18.6	74.0 ± 21.3 *	57.4 ± 21.4	<0.001
Brachial MAP (mmHg)	MAET	86.0 ± 6.1	93.0 ± 10.8 *	86.9 ± 6.3	<0.001
	RS	88.5 ± 5.6	98.5 ± 12.2 *	87.0 ± 8.1	<0.001
Brachial PP (mmHg)	MAET	63.2 ± 10.6	76.6 ± 17.7 *	60.9 ± 9.2	<0.001
	RS	65.4 ± 8.7	86.7 ± 19.3 *	60.4 ± 9.5	<0.001
Aortic PWV (m/s)	MAET	7.2 ± 1.4	7.0 ± 1.0	6.9 ± 2.4	0.159
	RS	7.5 ± 1.0	7.4 ± 1.3	7.0 ± 1.4	0.057
Carotid PWV (m/s)	MAET	8.0 ± 1.0	7.9 ± 1.0	8.7 ± 2.6	0.135
	RS	8.2 ± 1.2	8.0 ± 2.2	7.6 ± 1.1	0.070
Femoral PWV (m/s)	MAET	9.5 ± 1.3	8.9 ± 1.2 *	8.9 ± 1.2 *	0.022
	RS	9.6 ± 1.0	8.7 ± 1.1 *	8.5 ± 1.3 *	0.003

*, *p* < 0.05 significantly different from rest value; ^§^, *p*-value from ANOVA; MAET, maximal aerobic exercise test; MAP, mean arterial pressure; PP, pulse pressure; PWV, pulse wave velocity; RS, rescue simulation.

## Data Availability

All data relevant to the study are included in the article.

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
