# Peer review of "Comparison of Acute Arterial Responses Following a Rescue Simulation and Maximal Exercise in Professional Firefighters"

_healthcare, 2023, doi:10.3390/healthcare11071032_

Round 1
Reviewer 1 Report
First, I would like to thank the opportunity to review this manuscript.
Santos and colleagues presented an interesting study to compare hemodynamic and cardiorespiratory responses following rescue simulation and maximal exercise in firefighters.
A point-by-point questions/suggestions are presented below:
Abstract
1) The background presented in the abstract is too extensive (e.g., the last sentence before the aim of the study), and the study design is not presented.
2) Arterial stiffness is only presented as an acronym, never in full.
3) The conclusion does not reflect the entire results presented. No CPET results and comparisons are taken into account.
Introduction
1) The first citation presented is numbered 8. I think that is wrong, according to the instructions of the journal to submit manuscripts.
2) The authors used different terms to indicate CPET (e.g., maximal exercise, maximal aerobic exercise). Please homogenize in just one term.
3) In the sentence "Despite existing research describing the hemodynamic responses following exercise in a clinical setting among non-firefighter populations, there are a lack of data describing post exercise hemodynamic responses.", no references are presented. Please provide.
Material and Methods
Subjects
1) In my opinion this information should be presented in Results, along with a flow chart.
2) No inclusion and exclusion criteria are presented.
Study design
1) The study design presented is incorrect.
2) Please explain the instructions and interpretation of the PROMS: SF-36, IPAQ, and Framingham score. Psychometric properties are necessary.
Intervention sessions ( I am not sure if it is appropriate to consider the sessions as interventions)
1) How was assessed the Rating of Perceived Exertion (RPE)? Borg scale? please indicate.
2) No reference is mentioned for the volitional exhaustion criteria.
Central and Peripheral Arterial Stiffness Indices
1) The equations presented in this subsection are not presented according to the journal's instructions.
Dual-energy X-ray absorptiometry
1) "lean soft tissue" or lean mass?
Statistical Analysis
1) "Controlling PWV for MAP did not alter the results". Is unclear this information.
2) "Cardiorespiratory and hemodynamic variables were compared between CPET and RS conditions using independent t-test." Please explain the reasons for the use of this test.
Results
1) 18 smokers were included. Why were they included? Can't smoking habits influence the results?
2) in the legend of table 2 "*, Significantly different from rest", I do not understand this information along with the information presented in the respective table.
3) In the results of table 3, the p values adjusted are not presented. Do authors believe that this information is important?
4) What are the reasons for only presenting the VO2, MAP and HR for the comparison of acute cardiorespiratory and hemodynamic variables in maximal cardiopulmonary exercise test and rescue simulation conditions? Variables such RER, ventilation (and more) are also important and interesting to explore in results and discussion.
5) Results from F-36, IPAQ, and Framingham score are not presented.
Discussion/Conclusion
1) The conclusion of this manuscript did not fully reflect the discussion and the results presented. please reformulate.
Author Response
We appreciate all your comments, which are extremely relevant and enriching to our work.
Regarding the comments:
Introduction
1) The first citation presented is numbered 8. I think that is wrong, according to the instructions of the journal to submit manuscripts.
Authors: we apologize, it must have been a last minute change and we did not notice the change in references, they are all corrected and changed
2) The authors used different terms to indicate CPET (e.g., maximal exercise, maximal aerobic exercise). Please homogenize in just one term.
Authors: revised in all document
3) In the sentence "Despite existing research describing the hemodynamic responses following exercise in a clinical setting among non-firefighter populations, there are a lack of data describing post exercise hemodynamic responses.", no references are presented. Please provide.
Authors: done
Material and Methods
Subjects
- In my opinion this information should be presented in Results, along with a flow chart.
Authors: This content explains the selection of the final sample and does not include the sample’s results. Thus, this content has been maintained in the Methods section.
2) No inclusion and exclusion criteria are presented.
Authors: The inclusion criteria are described in lines 76-78 and highlighted in yellow in the main document
Study design
1) The study design presented is incorrect.
Authors:: The study design has been revised accordingly.
2) Please explain the instructions and interpretation of the PROMS: SF-36, IPAQ, and Framingham score. Psychometric properties are necessary.
Authors: The data from the questionnaires were not used in this article, they were collected as part of a larger study, but in this work these data were not included, so we removed this information to ensure that there is no confusion. The Framingham risk score calculation is a scoring system used to estimate the likelihood that a person will develop cardiovascular disease within a given time period, in this case 10 years. For our sample, the risk was calculated to be less than 2%. Information on the results can be found in Table 1.
Intervention sessions (I am not sure if it is appropriate to consider the sessions as interventions)
1) How was assessed the Rating of Perceived Exertion (RPE)? Borg scale? please indicate.
Authors: A description of the RPE scale and procedures are provided in the Maximal Exercise Section. On the Maximal Aerobic Exercise Testing the RPE was assessed each minute using a 6-20 borg scale. On the Rescue Simulation we asked to each firefigther provided their RPE (6-20 Borg scale) before started and immediately after the RS.
2) No reference is mentioned for the volitional exhaustion criteria.
Authors: included
Central and Peripheral Arterial Stiffness Indices
1) The equations presented in this subsection are not presented according to the journal's instructions.
Authors: In the instructions it says that equations can be inserted inline and that all terms used in an equation should be defined in the text, and all this has been taken into account. We do not understand what is not consistent with the instructions of the journal.
Dual-energy X-ray absorptiometry
1) "lean soft tissue" or lean mass?
Authors: It is lean soft tissue mass because we used without bone.
Statistical Analysis
1) "Controlling PWV for MAP did not alter the results". Is unclear this information.
Authors: This content has been omitted for clarity as we indicated that MAP served as a co-variate in the ANOVA model.
2) "Cardiorespiratory and hemodynamic variables were compared between CPET and RS conditions using independent t-test." Please explain the reasons for the use of this test.
Authors: The analyses of interventions over time for AS indices were examined via two-way ANOVA with two within factors. Paired (dependent) sample t-tests were used for post-hoc analysis.
Results
1) 18 smokers were included. Why were they included? Can't smoking habits influence the results?
Authors: We appreciate the Reviewer’s comment. Smoking may influence arterial stiffness. However, given that this study utilized a within subjects / cross-over design, each firefighter served as their own control, thus controlling for any confounding effects of smoking status. Future research should investigate any moderating effect of smoking status on hemodynamic outcomes among firefighters. Also smoking is fairly common among firefighters. Thus, it is ecologically valid to inlcude smokers in this study.
2) in the legend of table 2 "*, Significantly different from rest", I do not understand this information along with the information presented in the respective table.
Authors: The legend has been revised to indicate that there was a significant difference between conditions.
3) In the results of table 3, the p values adjusted are not presented. Do authors believe that this information is important?
Authors: Yes, we have represented only on the legend p > 0.05, so it was represented by *.
4) What are the reasons for only presenting the VO2, MAP and HR for the comparison of acute cardiorespiratory and hemodynamic variables in maximal cardiopulmonary exercise test and rescue simulation conditions? Variables such RER, ventilation (and more) are also important and interesting to explore in results and discussion.
Authors: The selected variables were utilized for analysis because they directly address the primary outcomes of this study. Other variables may be of interest, however their inclusion would increase the risk of Type I statistical error, therefore a more focused statistical analysis was utilized.
5) Results from F-36, IPAQ, and Framingham score are not presented.
Authors: Addressed in previous comment.
Discussion/Conclusion
1) The conclusion of this manuscript did not fully reflect the discussion and the results presented. please reformulate.
Authors: Reformulated.
We have attached the document with the changes in Track Changes

Reviewer 2 Report
1. The reviewer suggests the authors have a control group in the study. This will make the study a lot stronger.
2. The authors need to provide validation data or literature on carotid PWV. PWV is normally used in carotid-femoral data.
If carotid-femoral artery pulse is measured, why did the authors use Carotid PWV instead of c-fPWV?
Author Response
We appreciate all your comments, which are extremely relevant and enriching to our work.
Regarding the comments:
- The reviewer suggests the authors have a control group in the study. This will make the study a lot stronger.
Authors: Thank you for the reviewer's suggestion. However, in the present study, each of the subjects served as their own control by participating in each condition. Thus, this design may be considered a methodological strength.
- The authors need to provide validation data or literature on carotid PWV. PWV is normally used in carotid-femoral data. If carotid-femoral artery pulse is measured, why did the authors use Carotid PWV instead of c-fPWV?
Authors: Thank you for your comment. We appreciate your suggestion regarding validation data or literature on carotid pulse wave velocity (PWV). This method is commonly used and the theory and validity of the measurment is also well established (Harada et al, Hearrt and Vessels, 2002; Sugawara et al, Heart and Vessels, 2000; Vriz et al, Hypertension Res., 2027). Although we did not establish specific reliability of the measurement for this study, the reliability of the measurement in general is well established (Heffernan et al, Int. J Hypertens., 2013; Lefferts et al, Front. Physiol., 2014). We acknowledge that PWV is commonly measured using the carotid-femoral pathway, but recent studies have also shown that carotid PWV can provide valuable information regarding arterial stiffness. Moreover, we would like to highlight that in our study, we measured aortic PWV in addition to carotid PWV.
We have attached the document with the changes in Track Changes
